# Evaluation of Biomarkers of Severity in Patients with COVID-19 Infection

**DOI:** 10.3390/jcm10173775

**Published:** 2021-08-24

**Authors:** Akitaka Yamamoto, Hideo Wada, Yuhuko Ichikawa, Hikaru Mizuno, Masaki Tomida, Jun Masuda, Katsutoshi Makino, Shuji Kodama, Masamichi Yoshida, Shunsuke Fukui, Isao Moritani, Hidekazu Inoue, Katsuya Shiraki, Hideto Shimpo

**Affiliations:** 1Department of Emergency and Critical Care Center, Mie Prefectural General Medical Center, Yokkaichi 510-0885, Japan; akitaka-yamamoto@mie-gmc.jp (A.Y.); st25053@yahoo.co.jp (M.T.); 2Department of General and Laboratory Medicine, Mie Prefectural General Medical Center, Yokkaichi 510-0885, Japan; 3Department of Central Laboratory, Mie Prefectural General Medical Center, Yokkaichi 510-0885, Japan; ichi911239@yahoo.co.jp (Y.I.); solid.works.143@gmail.com (H.M.); 4Department of Cardiovascular Medicine, Mie Prefectural General Medical Center, Yokkaichi 510-0885, Japan; jun-masuda@mie-gmc.jp (J.M.); katsutoshi-makino@mie-gmc.jp (K.M.); 5Department of Respiratory Medicine, Mie Prefectural General Medical Center, Yokkaichi 510-0885, Japan; shuuji-kodama@mie-gmc.jp (S.K.); masamichi-yoshida@mie-gmc.jp (M.Y.); 6Department of Gastroenterology, Mie Prefectural General Medical Center, Yokkaichi 510-0885, Japan; m13092sf@jichi.ac.jp (S.F.); isao-moritani@mie-gmc.jp (I.M.); hidekazu-inoue@mie-gmc.jp (H.I.); katsuya-shiraki@mie-gmc.jp (K.S.); 7Mie Prefectural General Medical Center, Yokkaichi 510-0885, Japan; hideto-shimpo@mie-gmc.jp

**Keywords:** COVID-19, biomarkers, mild, moderate, severe

## Abstract

Object: Although many Japanese patients infected with coronavirus disease 2019 (COVID-19) only experience mild symptoms, in some cases a patient’s condition deteriorates, resulting in a poor outcome. This study examines the behavior of biomarkers in patients with mild to severe COVID-19. Methods: The disease severity of 152 COVID-19 patients was classified into mild, moderate I, moderate II, and severe, and the behavior of laboratory biomarkers was examined across these four disease stages. Results: The median age and male/female ratio increased with severity. The mortality rate was 12.5% in both moderate II and severe stages. Underlying diseases, which were not observed in 45% of mild stage patients, increased with severity. An ROC analysis showed that C-reactive protein (CRP), ferritin, procalcitonin (PCT), hemoglobin (Hb) A1c, albumin, and lactate dehydrogenase (LDH) levels were significantly useful for the differential diagnosis of mild/moderate I stage and moderate II/severe stage. In the severe stage, Hb levels, coagulation time, total protein, and albumin were significantly different on the day of worsening from those observed on the day of admission. The frequency of hemostatic biomarker abnormalities was high in the severe disease stage. Conclusion: The evaluation of severity is valuable, as the mortality rate was high in the moderate II and severe stages. The levels of CRP, ferritin, PCT, albumin, and LDH were useful markers of severity, and hemostatic abnormalities were frequently observed in patients in the severe disease stage.

## 1. Introduction

Since the first outbreak of coronavirus disease 2019 (COVID-19) in China [1,2], COVID-19 infections have spread worldwide, generating a pandemic [3]. The mortality rate of COVID-19 is approximately 2%, with 5–10% of patients developing severe and life-threating disease [3]. COVID-19 infection predominantly displays hyperinflammation and immune dysregulation, which induce multiorgan damage. The primary cause of mortality due to COVID-19 is severe acute respiratory distress caused by epithelial infection and alveolar macrophage activation in the lungs [4]. Therefore, immune modulation and suppression therapy may prevent the deterioration of the condition of COVID-19 patients [5]. Some COVID-19 patients develop severe disease or die [6], while many more develop mild or moderate disease [7]. The risk factors for severe disease are reported to include an age ≥65 years [8], malignant tumor [9], chronic obstructive pulmonary disease [10], chronic renal disease [11], diabetes mellitus [12], hypertension [13], hyperlipidemia [8], obesity [14], smoking [13], and immunodeficiency after transplantation [15]. Biomarkers including white blood cell (WBC) count, lymphocyte count, platelet count; albumin, ALT, lactate dehydrogenase (LDH), D-dimer, ferritin, interleukin-6, and procalcitonin (PCT) levels; and prothrombin time (PT) have been reported as specific biomarkers of severity [16]. In this study, specific biomarkers of severity and hemostatic abnormalities were examined in 152 patients with four disease stages of COVID-19 in Japan.

## 2. Materials and Methods

One hundred and fifty-two patients with COVID-19 infection (median age, 53.5 years; 25–75th percentile, 38.0–70.0 years; female, *n* = 78; male, *n* = 74) were admitted to Mie Prefectural General Medical Center from 1 April 2020 to 21 March 2021. The severity of COVID-19 was evaluated based on the Japanese Medical Care Guidelines [15]. There were 63 patients with mild disease (without pneumonia), 48 with moderate I disease (pneumonia without oxygen therapy), 32 with moderate II disease (pneumonia with oxygen therapy without mechanical ventilation), and 9 with severe disease (pneumonia with mechanical ventilation) (Table 1).

### 2.1. Measurement of Laboratory Biomarkers

Blood sampling was performed on the day of admission in hospitalized patients and throughout the clinical course in patients with severe disease. Total protein (TP), albumin, AST, lactate dehydrogenase (LDH), creatinine, and C-reactive protein (CRP) were measured by an Aqua-auto Kainos TP-II test, Aqua-auto Kainos ALB (Kainos, Tokyo, Japan), LaboFit AST, CicaFit LD-IFCC (Kanto Chemical Co., Inc., Tokyo, Japan), Signasu-auto CRE (Shino-test, Tokyo, Japan), and CRP-Latex X2 (Denka, Niigata, Japan), respectively, using a LABOSPECT006 (Hitachi High-Tech Co., Tokyo, Japan). PCT and ferritin were measured by Elecsys^®^ BRAHMS PCT and Elecsys^®^ Ferritin (Roche Diagnostics K.K., Tokyo, Japan), respectively, using a Cobas 8000 e602 (Roche Diagnostics K.K., Tokyo, Japan). The WBC count, hemoglobin (Hb), total neutrophil count, and total lymphocyte count were measured using a fully automatic blood cell counter XN-3000 (Sysmex Co., Kobe, Japan). HbA1c was measured using an Adams Hybrid AH 8280 (Arkray Factory, Shiga, Japan). The prothrombin time (PT), activated partial thromboplastin time (APTT), and D-dimer level were measured by a Thromborel S, Thrombocheck APTT-SLA, and LIASAUTO D-dimer Neo (Sysmex Co., Kobe, Japan), respectively, using an automatic coagulation analyzer CS-5100 (Sysmex Co., Kobe, Japan).

### 2.2. Statistical Analyses

The data are expressed as the median (25–75th percentiles). The difference in the frequency was analyzed by the chi-squared test. The significance of differences between groups was examined using the Mann-Whitney U test. *p*-Values < 0.05 were considered to indicate statistical significance. Cut-off values were determined by a receiver operating characteristic (ROC) analysis. A multivariate analysis with a stepwise regression test was performed. All statistical analyses were performed using Stat-Flex software (version 6; Artec Co. Ltd., Osaka, Japan).

## 3. Results

The median age and male/female ratio increased with severity (Table 1). The mortality rate was 12.5% in both the moderate II and severe stages. Comorbidity, which was not observed in 45% of mild stage patients, increased with severity (Table 1).

Hypertension, hyperlipidemia, or diabetes mellitus were observed ≥40% of patients with moderate II or severe disease. Heart failure, other pneumonia complications, and cerebrovascular accident were frequently observed among the patients who died within 30 days.

Regarding the laboratory data (Table 2), the neutrophil, lymphocyte, and platelet counts as well as the PT-INR and D-dimer levels in the moderate II and severe stages were significantly different from those in the mild and moderate I disease stages. The CRP, PCT, and ferritin levels in the moderate II and severe stages were significantly higher than those in the mild and moderate I disease stages. In the moderate II and severe stages the total protein and albumin levels, A/G ratio, and estimated glomerular filtration rate (eGFR) were significantly lower, and the levels of AST, LDH, and hemoglobin A1C levels were significantly higher in comparison with the mild and moderate I disease stages.

The ROC analysis of laboratory biomarkers in the mild and moderate I stages vs. moderate II and severe stages (Table 3, Appendix A), CRP (Figure 1), albumin, and A/G ratio showed ≥80.0% sensitivity; CRP, ferritin, albumin, A/G ratio, and hemoglobin A1c showed an odds ratio of ≥10.00; and CRP, PCT, ferritin, albumin, A/G ratio, and LDH showed an area under the curve (ARC) ≥0.800. In our multivariate analysis, CRP (*p* = 0.006) and AST (*p* = 0.033) were identified as significant factors in mild to moderate II stage vs. severe stage; albumin (*p* = 0.022), CRP (*p* = 0.001), and LDH (*p* = 0.011) were identified as significant factors in mild to moderate I stage vs. moderate II to severe stage; and LDH (*p* = 0.032) and CRP (*p* = 0.036) were identified as significant factors in mild stage vs. moderate I to severe disease stages.

In patients with severe disease, the neutrophil, hemoglobin, and platelet counts; PT-INR, APTT, TP, and albumin levels; and the A/G ratio on the day of each instance of worsening were significantly different from those on the day of admission (Table 4). The frequencies of hemostatic abnormalities were as follows: hemoglobin ≤ 10 g/dL, 44.4%; PT-INR ≥ 1.26, 44.4%; D-dimer ≥ 3.0 μg/mL, 44.4%; platelet count ≤ 10 × 10^10^/L, 55.5%; and APTT ≥ 50 s, 77.8% (Table 5).

## 4. Discussion

In the present study, the mortality rate was 0% in the mild and moderate I disease stages of COVID-19 infection, 12.5% in both moderate II and severe disease stages, and 3.3% overall, suggesting a mortality rate similar to other reports [3,16]. These findings suggest that four patients without mechanical ventilation died, and the direct cause of death was not COVID-19 in these patients. In addition to the management of severe COVID-19, the prevention of progression from mild/moderate I to moderate II and severe disease stages is important. Comorbidity, diabetes mellitus, hypertension, and cerebral vascular accident were related to the disease severity of COVID-19 infection. The elevation of Hb A1c and onset of hypertension may have occurred before the onset of COVID-19 infection, suggesting the importance of antihypertensive therapy and glycemic control before the onset of COVID-19 infection.

An increased WBC, especially increased neutrophil counts and decreased lymphocyte counts, was useful for predicting the disease severity of COVID-19. A decrease in lymphocytes due to apoptosis of CD4+ lymphocytes is correlated with the elevation of inflammatory cytokines [17]. CRP and ferritin were also useful biomarkers for predicting the disease severity of COVID-19. Ferritin elevation is believed to be caused by the cytokine storm and secondary hemophagocytic lympho-histiocytosis [18]. Elevation of CRP and ferritin and a reduction in lymphocytes may occur in association with a cytokine storm [19]. Although there was a large difference in the PCT level, this level was within normal range in the mild, moderate I, and moderate II disease stages. PCT is a specific biomarker for bacterial infection and is not markedly elevated in patients with common viral infections [20]. Therefore, PCT elevation may suggest bacterial co-infection or markedly severe inflammation in the severe disease stage of COVID-19 infection.

The change in AST was small and levels fell to within the normal range in patients with moderate II disease. LDH levels have been reported to be associated with Acute Physiology and Chronic Health Evaluation (APACHE) II, Sequential Organ Failure Assessment (SOFA), and computed tomography (CT) semiquantitative rating scores [21], and LDH is suggested to be a useful marker for multiple organ failure. TP and albumin levels, A/G ratio, and eGFR are also related to multiple organ failure. Thus, these biomarkers are associated with organ failure or renal function, indicating a correlation with the disease severity of COVID-19 infection.

Although D-dimer is a well-known biomarker of thrombosis in COVID-19 [22], the increase of D-dimer was small in the present study. These findings suggest that in the present study there were a few patients with severe disease with no resulting thrombotic complications. In the severe disease stage, Hb and platelet counts were significantly decreased, and the PT-INR and APTT were significantly prolonged, suggesting that the hemostatic abnormalities occurred after admission in patients with severe COVID-19. A high frequency of hemostatic abnormalities, including Hb, platelet count, PT-INR, and D-dimer abnormalities, suggests that a coagulopathy similar to thrombotic microangiopathy (TMA) [23] with disseminated intravascular coagulation (DIC) [24] may occur in patients with severe COVID-19. DIC and TMA produce similar pathological states and are associated with poor outcomes due to marked platelet activation [25]. Accordingly, aspirin treatment was reported to improve the mortality of patients with COVID-19 [26].

There are several limitations associated with the present study. Specifically, this study was observational and thus affected by comorbidities and varying therapies. Several patients died of old age or due to comorbidities rather than due to the severity of their COVID-19 infection. For most patients without severe disease, blood sampling was performed on the day of admission. In addition, the study population was relatively small, and there were few cases in the severe stage.

## 5. Conclusions

The mortality rate was high in COVID-19 patients with moderate II/severe disease. CRP, ferritin, PCT, albumin, and LDH levels were useful markers of severity, and hemostatic abnormalities were frequently observed in patients with severe disease.

## Figures and Tables

**Figure 1 jcm-10-03775-f001:**
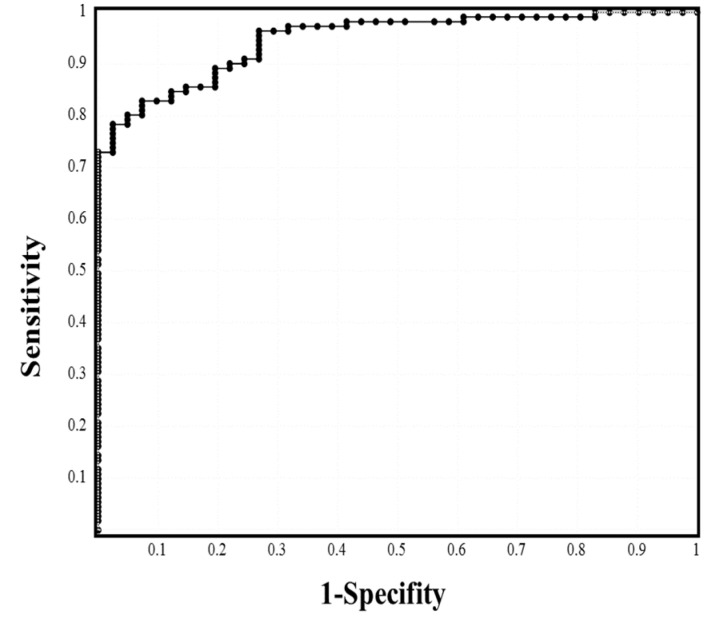
ROC analysis to determine the cutoff values of CRP (mild to moderate I vs. ≥moderate II to severe).

**Table 1 jcm-10-03775-t001:** Relationship between age, sex, mortality, and comorbidity and severity of COVID-19.

Stage	Mild	Moderate I	Moderate II	Severe
Number	63	48	32	9
Age, years (25–75th percentile)	41.0 (29.3–57.8)	50.5 * (38.0–64.0)	66.5 ***^, ###^ (58.5–80.5)	70.0 **^, ##^ (63.8–74.0)
Male (percent in stage)	25 (39.7%)	22 (45.8%)	19 (59.4%)	8 * (88.8%)
Death (mortality)	0	0	4 (12.5%)	1 (11.1%)
Died of old age	0	0	2 (6.3%)	0
Pregnant patient, *n* (%)	3 (4.8%)	0	0	0
Comorbidity	Patient number (percent) in each disease stage			
No comorbidity	28 (45.0%)	15 (31.2%)	3 **^, #^ (9.4%)	0 *
Hypertension	12 (19.4%)	10 (20.8%)	9 (28.1%)	5 * (55.6%)
Hyperlipidemia	6 (9.7%)	11 (22.9%)	6 (18.8%)	2 (22.2%)
Diabetes mellitus	6 (9.5%)	4 (8.3%)	11 **^, ##^ (34.4%)	5 **^, ##^ (55.6%)
Heart failure	2 (3.2%)	1 (2.1%)	3 (9.4%)	1 (11.1%)
Renal failure	2 (3.3%)	3 (6.3%)	5 (15.6%)	2 (22.2%)
Other pneumonia *	1 (1.6%)	1 (2.1%)	5 * (15.6%)	0 (0.0%)
Cerebrovascular accident	0	1 (2.1%)	5 ** (15.6%)	1 (11.1%)
Asthma	4 (6.3%)	5 (10.4%)	2 (6.3%)	0
Neurodegenerative diseases	5 (7.9%)	2 (4.2%)	4 (12.5%)	0
Digestive diseases	3 (4.8%)	2 (4.2%)	4 (12.5%)	1 (11.1%)
HIV infection	2 (3.2%)	1 (2.1%)	0	0
Autoimmune diseases	3 (4.8%)	2 (4.2%)	2 (6.2%)	0
Solid cancer	3 (4.8%)	0	0	0
Endocrine diseases	2 (3.2%)	2 (4.2%)	1 (3.1%)	0
Other diseases	11 (17.5%)	11 (22.9%)	13 (40.6%)	3 (33.3%)

Many comorbidities overlapped. HIV, human immunodeficiency virus; Mild, without pneumonia; Moderate I, pneumonia without oxygen therapy; Moderate II, pneumonia with oxygen therapy; Severe, pneumonia with mechanical ventilation. * *p* < 0.05; **, *p* < 0.01; ***, *p* < 0.001 in comparison with mild stage. ^#^, *p* < 0.05; ^##^, *p* < 0.01; ^###^, *p* < 0.001 in comparison with moderate I stage. * Although PCR was positive for COVID-19 in these patients, CT indicated other pneumonia, or their culture showed a bacterial or fungal infection.

**Table 2 jcm-10-03775-t002:** Laboratory data on the day of admission.

	Mild	Moderate I	Moderate II	Severe
WBC (×10^9^/L)	5.2 (4.2–6.1)	5.5 (4.0–6.7)	5.7 (4.0–7.6)	6.0 (5.4–8.0)
Neutrophil (×10^9^/L)	2.7 (1.7–3.3)	3.2 (1.9–4.1)	4.4 **^, ##^ (2.2–5.9)	4.8 ***^, ###^ (4.4–7.1)
Lymphocyte (×10^9^/L)	1.5 (1.1–1.7)	1.3 (1.0–1.7)	9.8 **^, #^ (0.6–1.3)	0.7 **^, ##^ (0.5–1.1)
Hemoglobin (g/dL)	14.4 (12.7–15.4)	14.3 (13.5–15.0)	13.4 (12.4–14.9)	14.6 (13.4–14.7)
Platelet (×10^10^/L)	21.7 (18.7–25.2)	22.0 (17.7–26.6)	19.5 * (14.5–23.2)	13.8 **^, ##^ (10.6–19.0)
PT-INR	0.96 (0.90–1.01)	0.98 (0.94–1.02)	1.02 ** (0.96–1.07)	1.02 **^, #^ (0.99–1.10)
APTT (sec)	32.0 (29.0–33.0)	32.0 (30.0–35.0)	32.0 (30.0–35.0)	33.0 (28.5–34.8)
D-dimer (μg/mL)	0.5 (0.5–0.7)	0.5 (0.5–0.9)	0.8 ***^, #^ (0.5–2.0)	1.2 ***^, ##^ (0.7–1.8)
CRP (mg/mL)	0.20 (0.05–0.65)	0.58 *** (0.31–2.67)	5.59 ***^, ###^ (2.69–8.11)	9.21 ***^, ###^ (7.53–13.0)
Procalcitonin (ng/mL)	0.05 (0.04–0.06)	0.06 * (0.05–0.08)	0.08 ***^, ##^ (0.07–0.10)	0.95 **^, ##^
Ferritin (ng/mL)	150 (83–342)	258 (115–369)	541 ***^, ###^ (378–1288)	1530 **^, ##^
Total protein (g/dL)	7.10 (6.88–7.50)	7.10 (6.80–7.45)	6.65 **^, #^ (6.30–7.25)	6.55 **^, #^ (6.30–6.95)
Albumin (g/dL)	4.20 (3.90–4.50)	4.00 * (3.80–4.30)	3.20 ***^, ###^ (3.00–3.70)	3.45 ***^, ###^ (2.80–3.55)
A/G ratio	1.40 (1.30–1.60)	1.30 ** (1.20–1.40)	1.00 ***^, ###^ (0.80–1.10)	1.05 ***^, ##^ (0.90–1.10)
T-Bil (mg/dL)	0.53 (0.37–0.68)	0.60 (0.46–0.77)	0.71 ***^, ##^ (0.65–1.09)	0.69 (0.50–0.98)
AST (U/L)	20.0 (16.3–24.0)	22.0 * (19.0–31.5)	35.0 ***^, ##^ (25.5–47.0)	43.0 ***^, ###^ (34.0–140)
LDH (U/L)	176 (155–195)	202 ** (175–249)	254 ***^, ##^ (216–333)	428 ***^, ###^ (318–731)
Creatinine (mg/dL)	0.67 (0.55–0.82)	0.69 (0.58–0.89)	0.76 (0.65–1.01)	1.24 * (0.68–1.51)
eGFR	84.5 (73.5–104)	79.5 (68.5–92.5)	65.0 ***^, #^ (54.3–84.3)	45.0 * (37.0–84.0)
Hemoglobin A1c (%)	5.60 (5.40–6.08)	5.80 (5.50–6.10)	6.45 ***^, ###^ (5.90–7.45)	6.70 ***^, ###^ (6.55–8.43)

WBC, white blood cell; PT-INR, prothrombin time—international normalized ratio; APTT, activated partial thromboplastin time; CRP, C reactive protein; A/G ratio, albumin/globulin ratio; T-Bil, total bilirubin; LDH, lactate dehydrogenase; eGFR, estimated glomerular filtration rate; Mild, without pneumonia; Moderate I, pneumonia without oxygen therapy; Moderate II, pneumonia with oxygen therapy; Severe, pneumonia with mechanical ventilation. *, *p* < 0.05; **, *p* < 0.01; ***, *p* < 0.001 in comparison with mild stage. ^#^, *p* < 0.05; ^##^, *p* < 0.01; ^###^, *p* < 0.001 in comparison with moderate I.

**Table 3 jcm-10-03775-t003:** ROC analysis to determine cutoff values of laboratory data (mild to moderate I vs. ≥moderate II to severe).

	Cutoff Value	Sensitivity	Odds Ratio	AUC
WBC	5.5 × 10^9^/L	55.0%	1.491	0.580
Neutrophil	3.5 × 10^9^/L	68.5%	5.067	0.734
Lymphocyte	1.1 × 10^9^/L	65.8%	3.990	0.695
Hemoglobin	14.0 g/dL	57.0%	1.898	0.580
Platelet	20.1 × 10^10^/L	59.3%	2.269	0.662
PT-INR	0.997	61.4%	2.557	0.676
APTT	32.8 s	53.2%	1.316	0.530
D-dimer	0.60 μg/mL	67.4%	4.438	0.697
CRP	2.36 mg/mL	85.0%	33.65	0.946
Procalcitonin	0.069 ng/mL	74.6%	9.387	0.853
Ferritin	379 ng/mL	78.3%	14.12	0.872
Total protein	6.89 g/dL	62.2%	2.558	0.703
Albumin	3.73 g/dL	81.7%	20.44	0.882
A/G ratio	1.10	84.6%	30.67	0.897
T-Bil	0.67 mg/dL	66.5%	4.248	0.711
AST	27.8 U/L	76.2%	11.86	0.794
LDH	222 U/L	76.9%	3.939	0.886
Creatinine	0.74 mg/dL	62.3%	3.163	0.630
eGFR	75.5	65.3%	3.571	0.690
Hemoglobin A1c	6.11%	73.9%	10.63	0.788

WBC, white blood cell; PT-INR, prothrombin time—international normalized ratio; APTT, activated partial thromboplastin time; CRP, C reactive protein; A/G ratio, albumin/globulin ratio; T-Bil, total bilirubin; LDH, lactate dehydrogenase; eGFR, estimated glomerular filtration rate; Mild, without pneumonia; Moderate I, pneumonia without oxygen therapy; Moderate II, pneumonia with oxygen therapy; Severe, pneumonia with mechanical ventilation. AUC, area under the curve.

**Table 4 jcm-10-03775-t004:** Laboratory data (data on admission vs. data at time of worsening) in patients with severe stage disease.

	Admission Data	Worsening Data	Worsening Period (Day)	*p*-Value
WBC (×10^9^/L)	6.0 (5.4–8.5)	18.6 (15.4–23.7)	10.0; 5.3–12.3	0.046
Neutrophile (×10^9^/L)	4.5 (3.8–5.2)	16.5 (13.1–21.1)	10.0; 6.0–11.5	0.044
Lymphocyte (×10^9^/L)	0.6 (0.2–0.9)	0.4 (0.2–0.5)	3.0; 1.0–7.3	0.229
Hemoglobin (g/dL)	14.6 (13.0–14.7)	10.2 (9.0–11.4)	13.0; 10.5–15.5	<0.001
Platelet (×10^10^/L)	13.8 (10.6–19.0)	7.3 (5.3–11.5)	1.0; 1.0–10.8	0.020
PT-INR	1.02 (0.99–1.11)	1.29 (1.17–1.37)	12.0; 3.8–12.5	0.006
APTT (sec)	33.0 (28.0–33.5)	81.5 (58.5–180)	2.0; 1.8–9.3	0.005
D-dimer (μg/mL)	1.2 (0.7–1.8)	3.7 (1.7–8.1)	6.0; 1.0–10.3	0.155
CRP (mg/mL)	9.21 (7.53–13.02)	11.67 (9.65–15.27)	2.0; 1.0–6.5	0.728
Total protein (g/dL)	6.55 (6.30–6.95)	4.20 (4.15–4.55)	7.0; 5.5–15.0	<0.001
Albumin (g/dL)	3.45 (2.80–3.55)	1.90 (1.75–2.15)	10.0; 6.5–15.0	<0.001
A/G ratio	1.05 (0.90–1.10)	0.80 (0.75–0.90)	3.0; 3.0–7.3	0.005
T-Bil (mg/dL)	0.69 (0.50–0.98)	1.20 (0.61–1.61)	8.0; 1.0–12.8	0.110
AST (U/L)	43.0 (34.0–139.5)	65.0 (52.0–119.5)	3.0; 1.0–9.0	0.900
LDH (U/L)	428 (317–731)	476 (395–723)	2.0; 1.0–7.8	0.803
Creatinine (mg/dL)	1.24 (0.68–1.51)	1.28 (0.75–1.51)	3.0; 1.0–4.3	0.879
eGFR	45.0 (36.5–75.0)	43.0 (36.0–75.0)	1.0; 1.0–3.3	0.155

WBC, white blood cell; PT-INR, prothrombin time—international normalized ratio; APTT, activated partial thromboplastin time; CRP, C reactive protein; A/G ratio, albumin/globulin ratio; T-Bil, total bilirubin; LDH, lactate dehydrogenase; eGFR, estimated glomerular filtration rate; Mild, without pneumonia; Moderate I, pneumonia without oxygen therapy; Moderate II, pneumonia with oxygen therapy; Severe, pneumonia with mechanical ventilation.

**Table 5 jcm-10-03775-t005:** Frequency of hemostatic abnormalities.

	Hemoglobin ≤ 10 g/dL	Platelet Count ≤ 10 × 10^10^/L	PT-INR ≥ 1.26	APTT ≥ 50 s	D-Dimer ≥ 3.0 μg/mL
Case (%)	4/9 (44.4%)	5/9 (55.5%)	4/9 (44.4%)	7/9 (77.8%)	5/9 (44.4%)

PT-INR, prothrombin time—international normalized ratio; APTT, activated partial thromboplastin time.

## Data Availability

The data presented in this study are available on request from the corresponding author. The data are not publicly available due to privacy restrictions.

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
