# Peer review of "Evaluation of Biomarkers of Severity in Patients with COVID-19 Infection"

_jcm, 2021, doi:10.3390/jcm10173775_

Round 1
Reviewer 1 Report
In this retrospective observational study, Dr. Yamamoto and colleagues investigated blood biomarker profiles at different stages of COVID-19 and found that some biomarkers were strongly associated with increased severity and could be useful in the clinical practice. Overall, this is a nicely conducted explorative observational study. However, I have some comments and suggestions to be addressed by the authors:
- Please (re)check the use of English (perhaps the authors need native English corrections). Some grammatical errors are found in the abstract and the text, and need corrections, for example:
- Line 21: remove comma after "although"
- Line 22: "...and leading to death" please rephrase. Also, maybe "deteriorates" is more appropriate than "worsens"?
- Line 24: maybe "into" is more appropriate than "as"?
- Line 102: should be "neutrophil"
- Before the sentence in line 50 about biomarkers, I think the authors need to spend 1-2 sentences explaining that COVID-19 severity is also important to assign appropriate (immunomodulatory) therapy. This has been discussed in this paper, which could be useful to be added (PMID: 34321903).
- Lines 54-55: The limitations of the study should be placed after the discussion. Please move those sentences there. The authors are also strongly suggested to explore other possible limitations of the study, including that this is an observational study rather than RCT, which is more likely to be influenced by other factors than the disease severity (e.g., comorbidities and therapies).
- Table 1 is not a good way to report the baseline characteristics of the patients. Please see other similar publications on how to report the data. Mainly, the authors need to separate everything based on several groups, perhaps male / female, and compare each parameters. The authors might need statistical analysis to see if the differences are significant or not.
- The fact that there were 63 patients in the mild stage was not reflected by Table 1. Please revise the table.
- Also, in the text, the authors said that there were 8 patients in severe stage, while in the table there were 9 patients. Please clarify.
- Please add the percentage of patients in each stage of COVID-19 in Table 1. It is currently not clear enough.
- In the title of table 1, there is "subjects". what does it mean? I think the authors also need to revise the title of that table because the current title is not good.
- Supplementary Table 1 as mentioned in line 61 could not be found anywhere in the submission system. Please clarify.
- The male:female ratio in Table 1 is also not clear. Are they absolute numbers or percentages?
- The percentages in Table 2 are not clear. What were they compared to? For example, 45% of what? total patients in each stage? Please clarify. I think it would be better to add the total number of patients in each stage on the first row.
- We also need the statistical analyses to compare those groups in table 2, whether the differences we saw are significant. Add the p-value or variance if ANOVA is used.
- Was the data in table 3 extracted at the same timepoint (during admission to the hospital)? The authors said that they also measured those biomarkers during the disease course in severe COVID-19. Please add those data in the table as well. I assume that the authors would already have the daily / weekly biomarkers data of severe patients as a part of the close monitoring in hospital or ICU?
- The description about the star and pound signs (* and #) in table 3 is missing. Please add what they referred to in the table caption underneath the table. I assume that they are statistical significance?
- Although Table 4 is enough to report the output of ROC analysis, it would be better to have the actual ROC curves. The authors can select the significant / important curves (perhaps the ones with high AUC) to be shown in the main manuscript and the rest can go to the supplementary file.
- The title of table 5 needs a clarification whether it was only for severe cases or all cases.
- Please also mark the biomarkers that reached statistical significance in Table 5.
- Probably this one is personal, but it was not clear why Table 6 was added and why only for hemostatic biomarkers? I checked the text and the reason for adding this table was not described there as well.
- I think the authors need to link their findings with hyperinflammation and cytokine storm phenomena observed in severe-critical COVID-19. This has also been discussed in this editorial (PMID: 34224330).
- Line 188: please add "...rate was high in COVID-19 patients with moderate..." to improve clarity.
Author Response
Comment 1
- Please (re)check the use of English (perhaps the authors need native English corrections). Some grammatical errors are found in the abstract and the text, and need corrections, for example:
- Line 21: remove comma after "although"
- Line 22: "...and leading to death" please rephrase. Also, maybe "deteriorates" is more appropriate than "worsens"?
- Line 24: maybe "into" is more appropriate than "as"? "into“?
- Line 102: should be "neutrophil"
Response 1: The English errors including your examples, in the manuscript have now been revised.
Comment 2
- Before the sentence in line 50 about biomarkers, I think the authors need to spend 1-2 sentences explaining that COVID-19 severity is also important to assign appropriate (immunomodulatory) therapy. This has been discussed in this paper, which could be useful to be added (PMID: 34321903).
Response 2: This paper has now been cited and the introduction of immunomodulatory therapy was described.
Comment 3
- Lines 54-55: The limitations of the study should be placed after the discussion. Please move those sentences there. The authors are also strongly suggested to explore other possible limitations of the study, including that this is an observational study rather than RCT, which is more likely to be influenced by other factors than the disease severity (e.g., comorbidities and therapies).
Response 3: The sentence concerning the limitation was moved to the end of the Discussion, and the Limitation section was revised in accordance with the reviewer’s comments.
Comment 4
- Table 1 is not a good way to report the baseline characteristics of the patients. Please see other similar publications on how to report the data. Mainly, the authors need to separate everything based on several groups, perhaps male / female, and compare each parameters. The authors might need statistical analysis to see if the differences are significant or not.
Response 4: Table 1 and Table 2 were combined and remade as Table 1. Statistical analyses were added for each parameter.
Comment 5
- The fact that there were 63 patients in the mild stage was not reflected by Table 1. Please revise the table.
Response 5: Table 1 and Table 2 were combined and revised as Table 1.
Comment 6
- Also, in the text, the authors said that there were 8 patients in severe stage, while in the table there were 9 patients. Please clarify.
Response 6: The number of patients has now been corrected to nine.
Comment 7
- Please add the percentage of patients in each stage of COVID-19 in Table 1. It is currently not clear enough.
Response 7: Table 1 and Table 2 were combined and remade as Table 1.
Comment 8
- In the title of table 1, there is "subjects". what does it mean? I think the authors also need to revise the title of that table because the current title is not good.
Response 8: The title of Table 1 was revised.
Comment 9
- Supplementary Table 1 as mentioned in line 61 could not be found anywhere in the submission system. Please clarify.
Response 9: The Supplementary Table has now been uploaded.
Comment 10
- The male:female ratio in Table 1 is also not clear. Are they absolute numbers or percentages?
Response 10: Table 1 and Table 2 were combined and remade as Table 1. The male : female ratio was revised.
Comment 11
- The percentages in Table 2 are not clear. What were they compared to? For example, 45% of what? total patients in each stage? Please clarify. I think it would be better to add the total number of patients in each stage on the first row.
Response 11: Table 1 and Table 2 were combined and remade as Table 1.
Comment 12
- We also need the statistical analyses to compare those groups in table 2, whether the differences we saw are significant. Add the p-value or variance if ANOVA is used.
Response 12: Table 1 and Table 2 were combined and remade as Table 1. Statistical analyses were added for each parameter.
Comment 13
Was the data in table 3 extracted at the same timepoint (during admission to the hospital)? The authors said that they also measured those biomarkers during the disease course in severe COVID-19. Please add those data in the table as well. I assume that the authors would already have the daily / weekly biomarkers data of severe patients as a part of the close monitoring in hospital or ICU?
Response 13: The laboratory data of patients with severe disease were already included in Table 5 (revised Table 4). Data from the “Worsening period” have now additionally been included in Table 5 (revised Table 4).
Comment 14
- The description about the star and pound signs (* and #) in table 3 is missing. Please add what they referred to in the table caption underneath the table. I assume that they are statistical significance?
Response 14: “*” and “#” indicated statistical significance. The legend of Table 3 (revised Table 2) has now been revised for clarity..
Comment 15
- Although Table 4 is enough to report the output of ROC analysis, it would be better to have the actual ROC curves. The authors can select the significant / important curves (perhaps the ones with high AUC) to be shown in the main manuscript and the rest can go to the supplementary file.
Response 15: Figure 1 and Supplementary Figure 1 showing the results of the ROC analysis were added.
Comment 16
- The title of table 5 needs a clarification whether it was only for severe cases or all cases. Please also mark the biomarkers that reached statistical significance in Table 5.
Response 16: The title of Table 5 (revised Table 4) has been revised. “P values” indicating the statistical significance were already shown.
Comment 17
- Probably this one is personal, but it was not clear why Table 6 was added and why only for hemostatic biomarkers? I checked the text and the reason for adding this table was not described there as well.
Response 17: We suspect that occult DIC or TMA may deteriorate the condition of patients with COVID-19. We have mentioned this in the Discussion.
Comment 18
- I think the authors need to link their findings with hyperinflammation and cytokine storm phenomena observed in severe-critical COVID-19. This has also been discussed in this editorial (PMID: 34224330).
Response 18 This article has now been cited and discussed.
Comment 19
- Line 188: please add "...rate was high in COVID-19 patients with moderate..." to improve clarity.
Response 19: Line 188 was revised in accordance with the suggestion from reviewer.

Reviewer 2 Report
The submitted manuscript presented a descriptive clinical research involving 152 COVID patients. According to the Japanese Medical Care Guidelines, the hospitalized patients were first divided into four groups: mild, moderate I, moderate II, and severe. The statistical analysis on laboratory tests showed that CRP, PCT, ferritin, albumin, A/G ratio, and LDH might serve as a biomarker for severity.
Major:
- L61. Supplement data is missing.
- L69. Many laboratory tests were ordered multiple times during hospitalization. It is not clear which test result is included in the statistical analysis (Table 3).
- Does COVID indeed cause biomarker changes?
- There are more diabetic patients in Moderate II and severe group (34.4-55.6% vs. 8.3-9.5%) than milder groups. Is the elevated A1c a precondition or a symptom caused by severe Covid infection? Since A1c usually changes slowly, I suspect the elevation A1c is a precondition instead of a consequence of COVID infection.
- Is LDH elevation a precondition (more heart failure patients in severe groups)? Table 5 showed that LDH did not change on the day of worsening.
- The authors need to provide more data to demonstrate that these biomarkers change as the COVID infection progresses.
- Line 133. The day of worsening was not adequately explained. The criteria for "worsening" should be listed.
Author Response
Thank you for your valuable and helpful comments. We have fully revised our manuscript in accordance with the reviewer" comments. The revised points were indicated red letters with yellow highlights.
Comment 1
- L61. Supplement data is missing.
Response 1: The Supplementary Table has now been uploaded.
Comment 2
- L69. Many laboratory tests were ordered multiple times during hospitalization. It is not clear which test result is included in the statistical analysis (Table 3).
Response 2: “Blood sampling was performed on the day of admission in hospitalized patients and throughout the clinical course in patients with severe disease.” was stated in the Material and Methods section. The legend of Table 3 (revised Table 2) defining the symbols used in the statistical analysis has now been revised.
Comment 3
- Does COVID indeed cause biomarker changes?
- There are more diabetic patients in Moderate II and severe group (34.4-55.6% vs. 8.3-9.5%) than milder groups. Is the elevated A1c a precondition or a symptom caused by severe Covid infection? Since A1c usually changes slowly, I suspect the elevation A1c is a precondition instead of a consequence of COVID infection.
- Is LDH elevation a precondition (more heart failure patients in severe groups)? Table 5 showed that LDH did not change on the day of worsening.
- The authors need to provide more data to demonstrate that these biomarkers change as the COVID infection progresses.
Response 3: The behavior of biomarkers was further discussed in accordance with the suggestion from reviewer.
- We agree with the reviewer’s comment, so HbA1c has now been deleted from the conclusion.
- The change in LDH was not as slow as that of HbA1C. Although there was a significant change in the LDH level, this change was not occurred before the onset of COVID-19 infection. Few patients with a severe disease condition showed deterioration.
- Table 5 was revised to Table 4.
Comment 4
- Line 133. The day of worsening was not adequately explained. The criteria for "worsening" should be listed.
Response 4: The text and Table 5 (revised Table 4) were revised.

Round 2
Reviewer 1 Report
Thank you for the responses to my previous comments. However, some issues remain and I will reiterate them below:
- The writing still needs some improvements. Not only the small grammar corrections but also the coherence and the flow of the introduction and discussion are lacking. The authors need to make sure that the sentences are well connected, which will make the manuscript easier to read. Perhaps a help from a professional native English scientific writer is needed to improve the readability and clarity of this manuscript.
- Please remove the recently added statement and the citation in lines 185-186. It is not relevant and I think the authors have misinterpreted my comment. I was actually suggesting the authors to discuss the link between disease severity and hyperinflammation / cytokine storm. This association has been exemplified by Hertanto et al. but I don't think it is necessary to cite the aforementioned editorial.
- Line 26: please remove "as"
- Line 91: please replace "statistic" with "test"
- I am not sure why there is "death" in Table 1 (next to "severe"). I agree that it is important to discuss mortality but it is currently redundant. Row 5 is also about mortality. Please revise.
- Also, not sure why being senile is considered a disease. Please clarify
- Instead of saying "Number (percent in stage)", I think the authors need to specify how many of the patients have comorbidities in each disease stage (and swap the position with no underlying disease). This would be more informative.
- Also, please differentiate between underlying disease and comorbidity. They are not interchangeable. I believe comorbidities would be more appropriate.
- I am wondering why this value "5 (55.6%)" in diabetes mellitus row (severe column) was not statistically significant. Are the authors sure about this?
- In the title of Table 2, the authors need to (re)clarify when was the lab sample taken. It could be at the admission but it is important to mention in the title.
- Please make sure that the abbreviations are used right after being introduced. For example, procalcitonin needs to be changed with PCT soon after its first use.
- I am not sure why there is this statement "PCT elevation may suggest complication by bacterial infection" in lines 183-184. Is it relevant? In the severe stage, the value was elevated more than 10x (Table 2), did this indicate bacterial infection? Any evidence? Blood culture maybe?
- The limitations can be improved and better elaborated. Acknowledging the shortcomings of the study is one of the key components of a good quality manuscript.
Author Response
Dear Reviewer 1,
Thank you for your valuable and helpful comments. We have fully responded to the comments from the reviewer. The points of revision are indicated red font and yellow highlighting. The following points were revised.
Sincerely yours,
Hideo Wada
Thank you for the responses to my previous comments. However, some issues remain and I will reiterate them below:
Comment 1
- The writing still needs some improvements. Not only the small grammar corrections but also the coherence and the flow of the introduction and discussion are lacking. The authors need to make sure that the sentences are well connected, which will make the manuscript easier to read. Perhaps a help from a professional native English scientific writer is needed to improve the readability and clarity of this manuscript.
Response 1: The introduction and Discussion have been fully rewritten and checked by a professional editor who is a native speaker of English.
Comment 2
- Please remove the recently added statement and the citation in lines 185-186. It is not relevant and I think the authors have misinterpreted my comment. I was actually suggesting the authors to discuss the link between disease severity and hyperinflammation / cytokine storm. This association has been exemplified by Hertanto et al. but I don't think it is necessary to cite the aforementioned editorial.
Response 2: This sentence and article were deleted and the Discussion was rewritten.
Comment 3
- Line 26: please remove "as"
Response 3: “as” was removed.
Comment 4
- Line 91: please replace "statistic" with "test"
Response 4: “chi-square statistic” was changed to “chi-squared test”.
Comment 5
- I am not sure why there is "death" in Table 1 (next to "severe"). I agree that it is important to discuss mortality but it is currently redundant. Row 5 is also about mortality. Please revise.
Response 5: “Death” was deleted. Table 1 was revised.
Comment 6
- Also, not sure why being senile is considered a disease. Please clarify
Response 6: “Senility” was changed “Died of old age”. Table 1 was revised.
Comment 7
- Instead of saying "Number (percent in stage)", I think the authors need to specify how many of the patients have comorbidities in each disease stage (and swap the position with no underlying disease). This would be more informative.
Response 7: “Number (percent in stage)” was changed to “Patient number (percent) in each disease stage”. Table 1 was revised.
Comment 8
- Also, please differentiate between underlying disease and comorbidity. They are not interchangeable. I believe comorbidities would be more appropriate.
Response 8: “underlying disease” was changed to “comorbidity”. Table 1 was revised.
Comment 9
- I am wondering why this value "5 (55.6%)" in diabetes mellitus row (severe column) was not statistically significant. Are the authors sure about this?
Response 9: “5 (55.6%)” was changed to “5** ## (55.6%)” following the reanalysis.
Comment 10
- In the title of Table 2, the authors need to (re)clarify when was the lab sample taken. It could be at the admission but it is important to mention in the title.
Response 10: Title of Table 2 was changed to “Table 2 Laboratory data on the day of admission”.
Comment 11
- Please make sure that the abbreviations are used right after being introduced. For example, procalcitonin needs to be changed with PCT soon after its first use.
Response 11: Procalcitonin was deleted in Line 79 and PCT was deleted in the legend of Table 2, 3, and 4.
Comment 12
- I am not sure why there is this statement "PCT elevation may suggest complication by bacterial infection" in lines 183-184. Is it relevant? In the severe stage, the value was elevated more than 10x (Table 2), did this indicate bacterial infection? Any evidence? Blood culture maybe?
Response 12: PCT is specific biomarker for bacterial infection and is not markedly elevated in patients with viral infection [20]. This sentence was added to the Discussion. “bacterial infection” was changed to “specific condition”.
[20]. Hamade B, Huang DT.: Procalcitonin: Where Are We Now? Crit Care Clin. 2020; 36: 23-40.
Comment 13
- The limitations can be improved and better elaborated. Acknowledging the shortcomings of the study is one of the key components of a good quality manuscript.
Response 13: The Limitations and Acknowledgments section were revised.

Reviewer 2 Report
L73. "Blood sampling was performed on the day of admission in hospitalized patients and throughout the clinical course in patients with severe disease."
Suppose CRP or WBC were tested multiple times throughout the clinical course, which value was used for Table2. For example, Patient A has WBC 3.0 on day 1, 4.0 on day 5, 3.5 on day 10. It is not clear which value is used to calculate values in table 2. If all test results were included in table 2, some patients might get WBC tested three times while others get test once. Patients who got tested three times would be over-presented in table 2. Please provide a more detailed method for table 2.
A1c is still in the abstract conclusion.
Author Response
Dear Reviewer 2,
Thank you for your valuable and helpful comments. We have fully responded to the comments from the reviewer. The points of revision are indicated with red font and yellow highlighting. The following points were revised.
Sincerely yours,
Hideo Wada
Comment 1
L73. "Blood sampling was performed on the day of admission in hospitalized patients and throughout the clinical course in patients with severe disease."
Suppose CRP or WBC were tested multiple times throughout the clinical course, which value was used for Table2. For example, Patient A has WBC 3.0 on day 1, 4.0 on day 5, 3.5 on day 10. It is not clear which value is used to calculate values in table 2. If all test results were included in table 2, some patients might get WBC tested three times while others get test once. Patients who got tested three times would be over-presented in table 2. Please provide a more detailed method for table 2.
Response 1: Line 108. “Regarding the laboratory data (Table 2)” was changed to “Regarding the laboratory data on the day of admission (Table 2)”. The title of Table 2 was changed to “Table 2. Laboratory data on the day of admission.
Comments 2
A1c is still in the abstract conclusion.
Response 2: HbA1C was deleted from the conclusion of the Abstract.

Round 3
Reviewer 1 Report
Thank you for once again addressing my comments. I only have some minor remarks to be corrected:
- Line 159: "as the behavior of HbA1C is slow ..." what does it mean? I never heard slow behavior of A1C before. Please rephrase.
- Line 156: please fix the erroneous spacing there.
- Line 161: how antihypertensive could impact HbA1C? Please explain.
- In Table 1, I am curious how could the authors assess "other pneumonia" in the presence of SARS-CoV-2 pneumonia? Please explain.
- Table 1: pregnancy is not a disease so it is not appropriate to include within comorbidity
- Please refrain from using "regarding (this / that)" repeatedly. It is kind of disturbing because they are not always in the correct place (sometimes they are not necessary). Please consult with the native writer on how to write a good quality essay / manuscript.
- Line 164: "a decreased lymphocyte counts" remove "a" or "s" in "counts".
- Line 165: "CD4+lymphocytes". space is needed in between.
- Line 173: "Therefore, PCT elevation may suggest a specific condition in severe disease stage of COVID-19 infection" replace "specific condition" with "bacterial co-infection"
- Line 184: please speculate on the possible cause of the low increase of D-dimer in this study and compare with other (big sampled) studies (observational or preferably RCT).
- Line 194: "Several limitation associated with the present study warrant mention" Are the authors sure that this sentence has been checked by the native writer? I doubt it.
- Line 198: "addmission" should be "admission"
- Again, for the third and final time, I would like to strongly suggest the authors to carefully read the manuscript, fix the potential unclarities, typos and remaining grammatical errors.
Author Response
Dear Reviewer 1,
Thank you for your valuable and helpful comments、again. We have fully responded to the comments from the reviewer. The points of revision are indicated red font and yellow highlighting. The following points were revised.
Sincerely yours,
Hideo Wada
Comments and Suggestions for Authors
Thank you for once again addressing my comments. I only have some minor remarks to be corrected:
Comment 1
- Line 159: "as the behavior of HbA1C is slow ..." what does it mean? I never heard slow behavior of A1C before. Please rephrase.
Response 1. As Reviewer 2 used this phrase, we included it in our response. To avoid confusion, we have now deleted “As the behavior of HbA1C is low” in our revision of Lines 159-160.
Comment 2
- Line 156: please fix the erroneous spacing there.
Response 2. The erroneous spacing was improved.
Comment 3
- Line 161: how antihypertensive could impact HbA1C? Please explain.
Response 3. This sentence has been revised.
Comment 4
- In Table 1, I am curious how could the authors assess "other pneumonia" in the presence of SARS-CoV-2 pneumonia? Please explain.
Response 4. We wrote not “SARS-CoV-2 pneumonia” but “COVID-19 infection”. Although PCR was positive for COVID-19 in these patients, CT indicated other pneumonia, or their culture showed bacterial or fungal infection. This has now explained in the legend of Table 1.
Comment 5
- Table 1: pregnancy is not a disease so it is not appropriate to include within comorbidity
Response 5. Table 1 has been revised.
Comment 6
- Please refrain from using "regarding (this / that)" repeatedly. It is kind of disturbing because they are not always in the correct place (sometimes they are not necessary). Please consult with the native writer on how to write a good quality essay / manuscript.
Response 6. We have now had these points revised by a native speaker.
Comment 7
- Line 164: "a decreased lymphocyte counts" remove "a" or "s" in "counts".
Response 7. We have removed “a”, as this is a plural noun.
Comment 8
- Line 165: "CD4+lymphocytes". space is needed in between.
Response 8.
A space has been added.
Comment 9
- Line 173: "Therefore, PCT elevation may suggest a specific condition in severe disease stage of COVID-19 infection" replace "specific condition" with "bacterial co-infection"
Response 9. "Specific condition" has been replaced with “bacterial co-infection or markedly severe inflammation”.
Comment 10
- Line 184: please speculate on the possible cause of the low increase of D-dimer in this study and compare with other (big sampled) studies (observational or preferably RCT).
Response 10. Our speculation has been added.
Comment 11
- Line 194: "Several limitation associated with the present study warrant mention" Are the authors sure that this sentence has been checked by the native writer? I doubt it.
Response 11. This sentence has been rechecked by our native English-speaking medical editor, who has confirmed that the corrected version with “Limitations” is now adequate.
Comment 12
- Line 198: "addmission" should be "admission"
Response 12. "Addmission" has been changed to "admission"
Comment 13
- Again, for the third and final time, I would like to strongly suggest the authors to carefully read the manuscript, fix the potential unclarities, typos and remaining grammatical errors.
Response 13. As suggested, we have now had a professional medical editor whose native language is English recheck the revised manuscript.

Reviewer 2 Report
Thanks for the updates and congratulations for the great work.
Author Response
Thank you very much for your kind and helpful comments
